# Impact of preoperative magnetic resonance imaging on surgery and eligibility for intraoperative radiotherapy in early breast cancer

Wai Yee Chan[1], Wai Keong Cheah[1], Marlina Tanty Ramli Hamid ⓘ [1,2]*, Mohammad Nazri Md Shah[1], Farhana Fadzli[1], Shaleen Kaur[1], Mee Hoong See[3], Nur Aishah Mohd Taib[3], Kartini Rahmat[1]*

1 Department of Biomedical Imaging, Faculty of Medicine, Universiti Malaya, Kuala Lumpur, Malaysia,
2 Department of Radiology, Faculty of Medicine, University Teknologi MARA, Sungai Buloh, Selangor, Malaysia, 3 Department of Surgery, Faculty of Medicine, Universiti Malaya, Kuala Lumpur, Malaysia

* marlina352@uitm.edu.my (MTRH); katt_xr2000@yahoo.com (KR)

**Data Availability Statement:** All relevant data are within the manuscript and its Supporting Information files.

## Abstract

We looked at the usefulness of magnetic resonance imaging (MRI) in decision-making and surgical management of patients selected for intraoperative radiotherapy (IORT). We also compared lesion size measurements in different modalities (ultrasound (US), mammogram (MMG), MRI) against pathological size as the gold standard. 63 patients eligible for IORT based on clinical and imaging criteria over a 34-month period were enrolled. All had MMG and US, while 42 had additional preoperative MRI for locoregional preoperative staging. Imaging findings and pathological size concordances were analysed across the three modalities. MRI changed the surgical management of 5 patients (11.9%) whereby breast-conserving surgery (BCS) and IORT was cancelled due to detection of satellite lesion, tumor size exceeding 30mm and detection of axillary nodal metastases. Ten of 42 patients (23.8%) who underwent preoperative MRI were subjected to additional external beam radio-therapy (EBRT); 7 due to lymphovascular invasion (LVI), 2 due to involved margins, and 1 due to axillary lymph node metastatic carcinoma detected in the surgical specimen. Five of 21 (23.8%) patients without prior MRI were subjected to additional EBRT post-surgery; 3 had LVI and 2 had involved margins. The rest underwent BCS and IORT as planned. MRI and MMG show better imaging-pathological size correlation. Significant increase in the mean 'waiting time' were seen in the MRI group (34.1 days) compared to the conventional imaging group (24.4 days). MRI is a useful adjunct to conventional imaging and impacts decision making in IORT. It is also the best imaging modality to determine the actual tumour size.

## Introduction

Although earlier studies on BCS versus mastectomy showed similar survival outcomes [1–3], the local recurrence rates in patients who underwent BCS were reportedly higher [3, 4], despite the use of conventional MMG and US for surgical planning. Recent literature, however, have

**Funding:** This research was funded by the University Malaya Research Fund Assistance grant (BKP) (Grant number - BK006-2018, Principal Investigator - Dr Chan Wai Yee) and University Malaya Faculty Research Grant (Grant number - GPF009C-2019, Principal Investigator - Associate Professor Dr Faizatul Izza Binti Rozalli). The funders had no role in study design, data collection and analysis, decision to publish, or preparation of the manuscript.

**Competing interests:** The authors have declared that no competing interests exist.

shown that BCS and mastectomy had similar local recurrence and survival rates [1, 5], with local irradiation an important factor leading to reduced recurrence rate in patients who underwent BCS [2].

In patients with early breast cancer, BCS followed by fractionated EBRT is the current standard of care [6]. Increased use of BCS in the treatment of localised breast cancer is seen in the past few decades [7]. IORT, which is radiation given intraoperatively immediately upon the removal of the breast tumour, was first introduced in the 1960s and has recently regained popularity in the past two decades as an alternative to EBRT following BCS [7, 8]. Advantages of IORT include direct target tissue visualisation, lower radiation to nearby normal tissues, reduced overall cost of treatment, as well as improvement in the quality of life [8–11]. Similar rates of complications were seen in IORT and EBRT, with no significant difference in local recurrence [12]. In University Malaya Medical Centre (UMMC), IORT is used for risk-adapted patients with low-risk features (Targeted intraoperative radiotherapy [TARGIT] A criteria) while higher-risk patients are recruited under the TARGIT B trial [13].

Optimal preoperative assessment is much needed to ensure successful BCS to reduce the cost and psychosocial impact of re-operations. Conventional imaging of the breast for surgical decision making comprises of both MMG and US. MRI is not a routine assessment tool for patients with breast cancer. Indications for breast MRI include screening in high-risk patients, preoperative evaluation of multifocal-multicentric or bilateral breast cancer, assessment of treatment response of neoadjuvant chemotherapy (NAC), evaluation of patients post BCS to differentiate scar tissue from local recurrence and assessment of patients with metastatic axillary lymphadenopathy with no known primary malignancy [14]. In newly diagnosed breast cancer, MRI breast is used to define the extent of cancer, detect the presence of satellite lesions or additional disease in patients with dense breasts and identify primary cancer in patients with axillary nodal involvement [15].

In the past decade, there has been an increased use of preoperative breast MRI globally, particularly for assessment of eligibility of early breast cancer patients for BCS and IORT [16]. Promising results of breast MRI in detecting multifocal-multicentric diseases have been demonstrated with several studies reporting detection rates of secondary lesions of between 7–10% which were missed on conventional imaging in patients initially eligible for IORT, leading to change in surgical management and reduced recurrence rate, proving MRI to be a useful adjunct to conventional imaging [9, 17, 18]. Advantages of breast MRI include an accurate definition of the extent of cancer to aid in the prevention of re-excision of tumours with positive margins, assist in preoperative staging of these patients [19] as well as identify multifocal-multicentric and contralateral breast diseases [20]. It has also shown higher accuracy in determining tumour size compared to US and MMG [21–24], with the highest imaging-pathological concordance [20] and good concordance for lesions smaller than 20mm [25].

Although many recent studies favour the use of MRI in early breast cancer patients, there are some disadvantages of performing a preoperative MRI. Breast MRI has been associated with the detection of insignificant additional breast lesions, with a study reporting only 25% of biopsied lesions as malignant on histopathological examination (HPE) [26]. Due to these additional lesions detected on MRI, additional workup (imaging and biopsies) would be required to confirm their nature, leading to increase in significant added cost and 'wait time' as defined as the time from the diagnostic biopsy to the time of first surgical treatment, ranging from 11 to 22.4 days delay in the MRI group [27–30]. Some studies have also shown the association of MRI with an increased rate of mastectomy as the initial surgical therapy and positive margins [27, 28], with an increase in 1.8-fold in odds ratio compared to the group without MRI [27]. Two studies reported an increased rate of conversion to mastectomy post breast MRI (4.9% and 8.3%) [22, 31], while a third study did not [27]. These conversions were later deemed to be

appropriate based on the HPE results [22, 31]. Given the many added benefits and limitations of adding MRI into the preoperative assessment repertoire, we sought to evaluate the usefulness of pre-operative breast MRI in detecting multifocal, multicentric, and contralateral synchronous breast cancers compared to conventional breast imaging and its impact on IORT eligibility. We also aim to correlate the accuracy of lesion size measurements on MMG, US and MRI as well as compare the 'wait time' between the two groups.

## Materials and methods

### Study design

This was a cross-sectional study involving 63 patients with early breast cancer and were deemed eligible for BCS and IORT from March 2016 to December 2019 in UMMC. The study was conducted in adherence with the approved guidelines from the Medical Ethics Committee of University of Malaya Medical Centre (MREDIC No: 2018421–6235). Written informed consent were obtained.

### Patient selection

Patients with a unifocal invasive ductal carcinoma of tumour size less than 30mm on conventional imaging with no lymph node or metastatic disease. All patients, except for those requiring neoadjuvant chemotherapy and with MRI contraindication, were offered MRI. Neoadjuvant chemotherapy cases were excluded. The patients were divided into two groups, those who only had pre-operative conventional imaging (MMG and US) versus those with conventional imaging plus MRI.

### Equipment & techniques

All patients had full-field digital mammography (FFDM) in standard craniocaudal (CC) and mediolateral oblique (MLO) views with digital breast tomosynthesis (DBT) (Selenia Dimensions, Hologic, Bedford, Massachusetts, USA). Supplementary breast and axillary US was performed using diagnostic B-mode greyscale and colour US system (Philips iU22; Philips Healthcare, Bothell, Washington, USA) with a high frequency (12.5 MHz) linear transducer probe.

Breast MRI was performed in a 3.0 Tesla MAGNETOM Prisma® scanner (Siemens Healthcare, Munich, Germany) with a dedicated 18-channel breast coil for radiofrequency signal transmission and reception. The standard imaging protocol were T2W fast spin echo (echo time [TE] = 412ms, repetition time [TR] = 3200ms, a 256 x 256 matrix, field of view [FOV] of 250 x 250mm, slice thickness of 5mm, acquisition time of 6min 35sec), turbo inversion recovery magnitude [TIRM], diffusion-weighted imaging [DWI] (TE = 77ms, TR = 3400ms, 30 diffusion directions at $b = 0s/mm^2$ and $b = 1000s/mm^2$, 80 x 80 matrix, FOV of 240 x 240mm and slice thickness of 3mm, acquisition time of 5min 59sec) as well as dynamic contrast-enhanced T1W images in axial projections.

### Data collection and analysis

Patients' characteristics and HPE results were obtained from the electronic medical records (EMR). Images were retrieved from the local picture archiving and communication system (PACS) and reviewed by three board-certified radiologists (KR, MTRH, and WYC with 5–10 experience in breast imaging). Tumour size on MMG was measured by MTRH and WYC by consensus reading. Lesions on digitised external MMG images that were not measurable were excluded. The index tumour was measured in 3 dimensions on MRI. The largest dimension of the tumour in all 3 imaging modalities was taken as the tumour size, and compared to HPE.

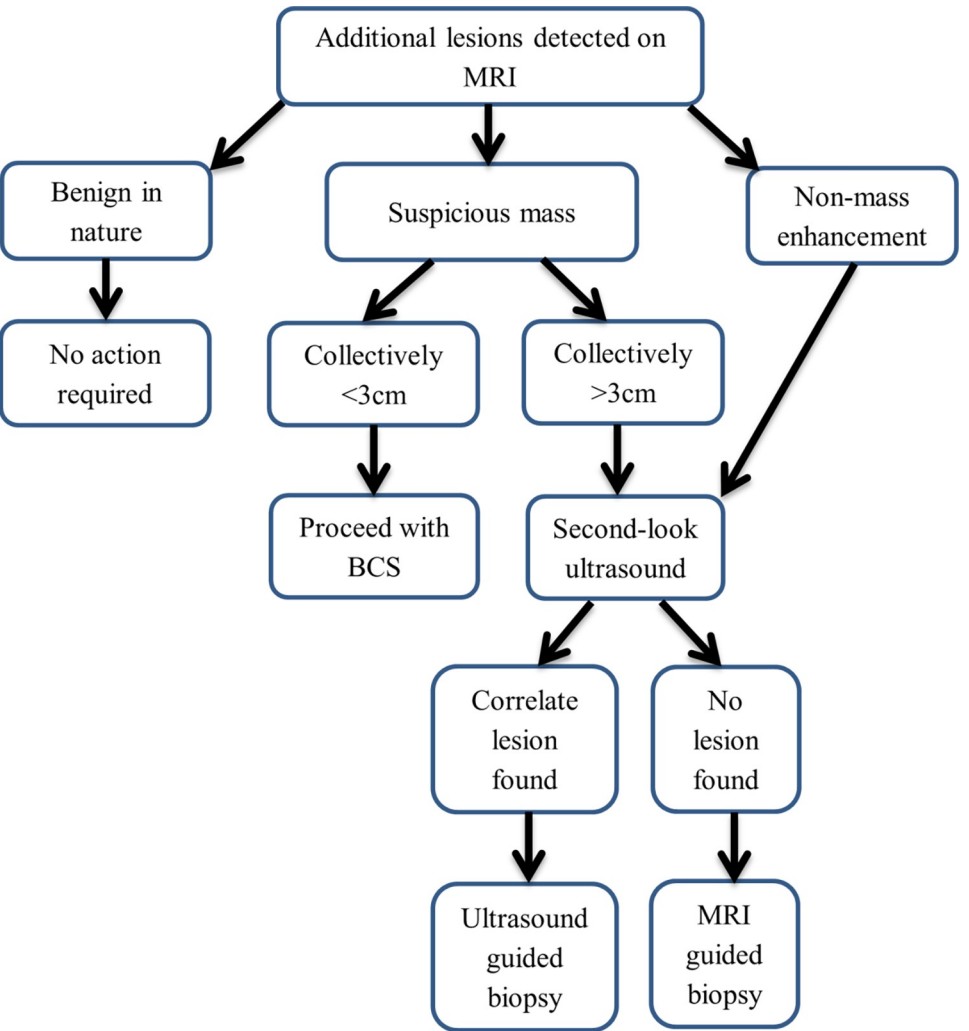

**Fig 1. Workflow for patients with additional lesions detected on MRI.**

Additional lesions detected on MRI were classified according to ACR BI-RADS 2015 lexicon into benign and suspicious masses, and non-mass enhancement (NME). For masses, these include assessment of lesions' shape, margin and internal enhancement characteristic. NMEs were assessed according to the distribution and internal enhancement pattern.

Multifocal disease is the presence of another focus/foci of cancer (satellite lesion) in the same quadrant of the index tumour, which collectively measures <3cm, whilst multicentric disease means presence of cancer foci in more than one quadrant of the breast. The management of these additional lesions were decided in a multidisciplinary team discussion with the surgical team as per the workflow in Fig 1.

A list of all the patients with accompanying data has been supplied as a S1 Data.

## Pathology

On gross examination, tumours larger than 10mm were measured using a ruler, whilst a microscope was used for lesions smaller than 10mm. As per recommendation, the definition of clear margin was no ink on invasive tumours [32] and at least 2 mm away from the inked

margin for ductal carcinoma in-situ (DCIS) [33]. The pathological size was taken as gold standard. Imaging measurement was compared with pathology size. Concordance between imaging and pathology was defined as size difference equal or less than 5mm. Underestimation or overestimation was defined as a discrepancy of measurement of more than 5mm.

### Statistical analysis

All data were entered into IBM SPSS Statistics Data Editor Version 25 for statistical analysis. Measurable variables were analysed and summarised using means, medians, standard deviations (SD), standard error of the mean (SEM), and ranges, while categorical variables were measured in sums and percentages. The primary evaluation criterion was the incidence of satellite lesions.

The mean tumour size on US, MMG, MRI, and HPE were compared and analysed with the Paired Samples T-Test. The patients were divided into two groups (imaging size $\leq$ 20mm, and > 20mm) and the same test was also used for comparison of tumour sizes between these two groups.

Sensitivity and specificity was calculated with the use of true positive and false negative. True positives were satellite lesions correctly diagnosed on imaging, whereas false negatives denote satellite lesions inaccurately diagnosed as unifocal disease [34].

## Results

### Patient population demographics and breast MRI findings

Of the 63 patients, 21 had conventional imaging only and 42 had additional MRI. The majority had BIRADS breast density B (n = 26, 41.3%), followed by BIRADS density C (n = 23, 36.5%). 5 (7.9%) had extremely dense breast which lowers the sensitivity of MMG. 7 tumours were missed on MMG but detected on supplementary US which were performed due to either heterogeneously or extremely dense breast parenchyma.

All patient had invasive ductal carcinoma on HPE. Most (83.3%) were noted to be oestrogen receptor/progesterone receptor (ER/PR) positive. LVI was seen in 30.2% of patients with the majority of tumours classified as Grade 1 and 2 based on the Modified Bloom and Richardson score. Detailed characteristics of the tumour in our study were summarised in Table 1, as well as molecular subtyping in Table 2.

All 42 index tumours were visible on MRI. Twenty-five additional lesions were detected on MRI; 16 focal lesions, 6 non-mass enhancements (NMEs), and 3 suspicious axillary lymph nodes. Malignancy was detected in 28% of these MRIs, which comprised 5 focal lesions (DCIS), 1 NME (DCIS), and 1 axillary lymph node (metastatic disease) making up 16.7% of patients with additional malignancy in the MRI group. As a result, there were increased numbers of second-look US as well as confirmatory biopsies (5 US-guided, 2 MRI-guided breast lesion biopsies as well as 2 US-guided axillary lymph node biopsies) compared to conventional imaging group. These findings are summarised in Fig 2. There was a significant (P<0.001) increase in the mean 'waiting time' in the MRI group (34.1 days, 95% CI = 29.9–38.3) compared to the conventional imaging group (24.4 days, 95% CI = 19.5–29.4).

### Impact on surgical management

In the conventional imaging cohort (n = 21), 5 patients (23.8%) had EBRT added due to involved margins (2 patients, both had underestimated tumour size on US and LVI (3 patients). Due to the involved margins, these 2 patients (9.5% of total patients) were subjected to secondary operation for margin clearance, while the patients with LVI were subjected to post-op EBRT only.

**Table 1. Tumour characteristics.**

| Characteristics | N | n | % |
|---|---|---|---|
| **Laterality** | | | |
| Left | 30/63 | | 47.6 |
| Right | 33/63 | | 52.4 |
| **Hormonal receptor status*** | | | |
| ER+/PR+ | 50/60 | | 83.3 |
| ER+/PR- | 5/60 | | 8.3 |
| ER-/PR+ | 1/60 | | 1.7 |
| ER-/PR- | 4/60 | | 6.7 |
| HER-2-neu+ | 5/60 | | 8.3 |
| **Lymphovascular invasion (LVI)**** | 19/63 | | 30.2 |
| **Grade** | | | |
| 1 | 15/63 | | 23.8 |
| 2 | 36/63 | | 57.1 |
| 3 | 10/63 | | 15.9 |
| Unknown | 2/63 | | 3.2 |

*2 patients had surgery done in other centres, 1 patient is still undergoing neoadjuvant chemotherapy

In the MRI group (n = 42), 10 patients (23.8%) had additional EBRT due to post-surgically proven LVI (7 patients), involved margins (2 patients, both had underestimated tumour size on MRI) and metastatic axillary lymph nodes (1 patient). Both patients (4.8% of total patients) had a change in management, in which one had additional EBRT while the other proceeded with a completion mastectomy. The risk of margin involvement doubled in patients without MRI (9.5% in the conventional group versus 4.8% in the MRI group), although there was no statistical significance (Chi Square, p = 0.465). In total, 15 patients (5 in the conventional imaging and 10 in the MRI group respectively) were subjected to additional EBRT.

5 patients (11.9%) in the MRI group were not eligible for breast-conserving surgery and IORT, 2 of whom had a multifocal-multicentric disease and converted to mastectomy and 2 with tumours larger than 30mm on MRI (tumour sizes were underestimated on conventional imaging). The 5th patient with axillary lymph node metastasis only detected on MRI had additional EBRT. One patient with tumour size larger on MRI was subjected to mastectomy while the other had NAC before surgery. The remaining MRI patients (n = 27) underwent BCS and IORT as planned (Fig 3).

Figs 4–7 illustrate cases in which MRI demonstrated additional tumour findings as well as tumour size estimation.

**Table 2. Frequency of molecular subtypes in the study population.**

| Subtype | N | % |
|---|---|---|
| Luminal A | 46/54 | 85.1 |
| Luminal B | 5/54 | 9.2 |
| HER2+ | 2/54 | 3.9 |
| Triple Negative Breast Cancer | 1/54 | 1.8 |

*9 patients did not have complete hormonal information in order to fit the above subtypes

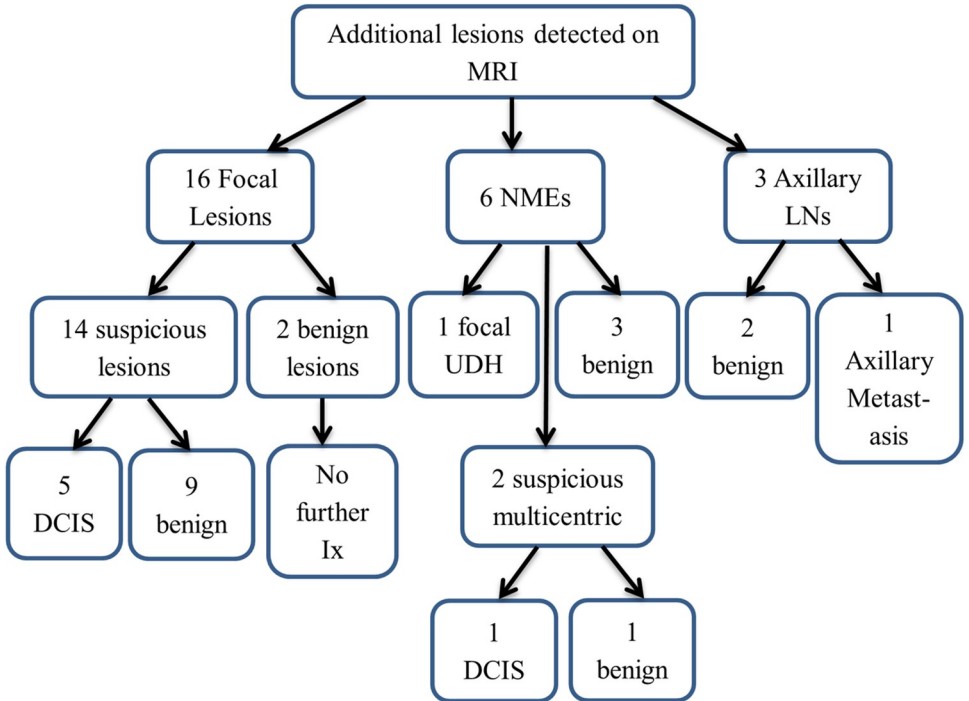

**Fig 2. Additional lesions detected on MRI.** NME = Non-mass enhancement, LNs = Lymph nodes, UDH = Usual ductal hyperplasia, DCIS = Ductal carcinoma in-situ, Ix = Investigation.

## Statistical analysis and imaging-pathological size concordance

The positive and negative findings are summarized in Table 3. A total of 60, 29, and 39 index tumours on US, MMG, and MRI respectively were included. 3 patients were excluded; 2 had

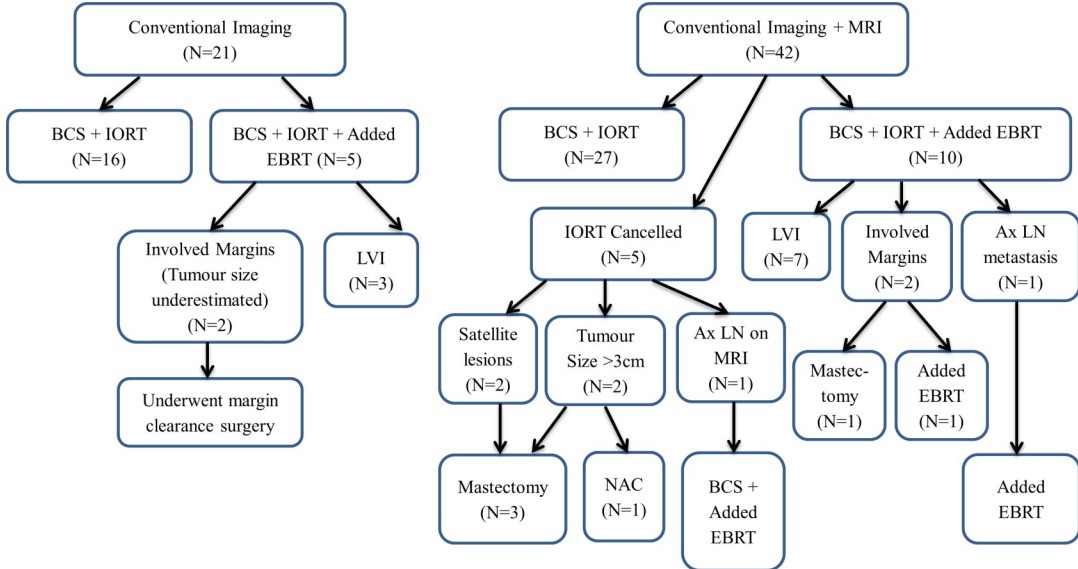

**Fig 3. Flowchart outcome for conventional imaging only and conventional imaging plus MRI cohorts.** (BCS = Breast-Conserving Surgery, IORT = Intraoperative Radiotherapy, EBRT = External Beam Radiotherapy, LVI = Lymphovascular invasion, Ax LN = Axillary lymph node, NAC = Neoadjuvant chemotherapy).

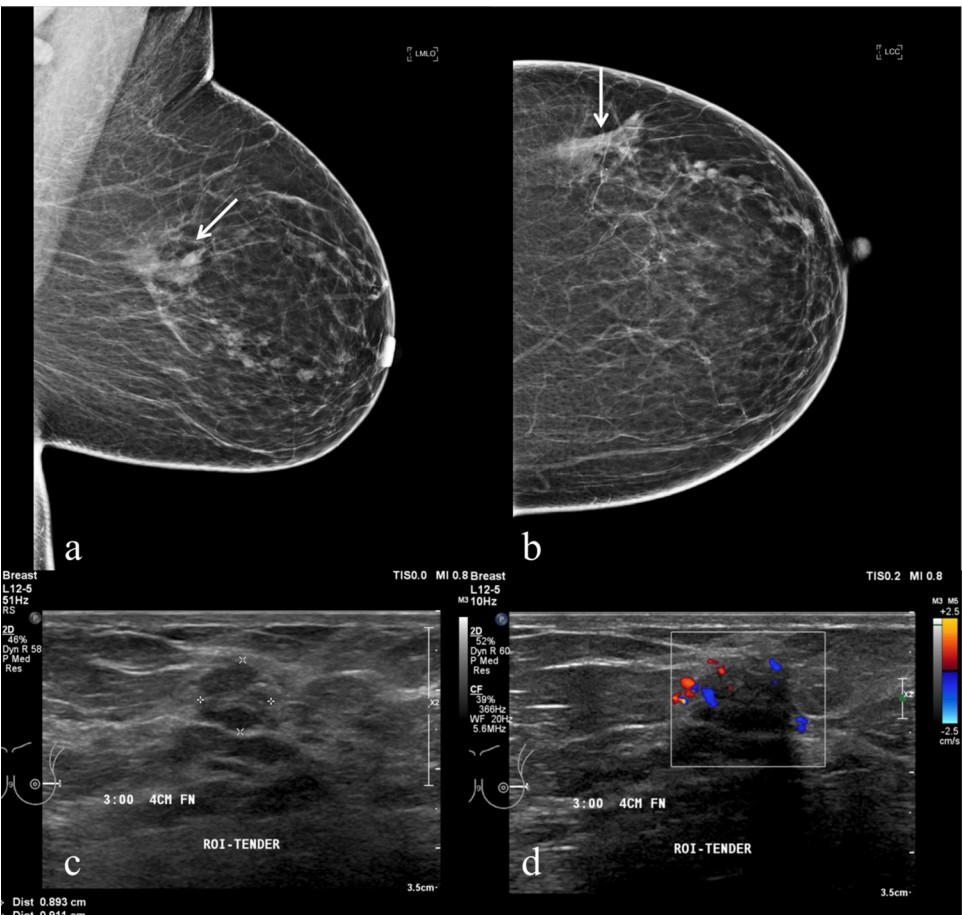

**Fig 4. A 49-year-old woman with invasive carcinoma who subsequently underwent mastectomy due to imaging discordance.** Left MMG in (A) MLO and (B) CC views showing a spiculated lesion (thin arrow) in the mid outer region with foci of microcalcifications on a background of fatty breast parenchymal pattern (BIRADS density A). Corresponding US in (C) transverse view showing an ill-defined hypoechoic mass in the left 3 o'clock position, measuring 9 x 9mm with the presence of internal vascularity (D).

surgeries in other centres and 1 was currently undergoing NAC and awaiting surgery. For MMG, only 29 patients were included as the rest had hard/soft copy external MMG images without measurement. Tumor size distribution and comparison are shown in Tables 4 and 5 and Fig 8.

Discordance was observed in larger sized tumours (>20mm) and oestrogen receptor/progesterone receptor (ER/PR) negative tumours (3 of 4 patients). Chi-Square Tests done showed no statistically significant association between LVI or breast density and size discordance.

## Discussion

The increased use of BCS as a treatment of localised breast cancer created a need for optimum assessment of the extent of cancer as well as detecting multifocal-multicentric disease [35]. Failure to detect multifocal-multicentric disease, treatable by radio and chemotherapy, may lead to local recurrence [35]. Detection of contralateral cancer is also of utmost importance as these lesions will not be covered by radiotherapy [36]. MRI has been proven to be the most sensitive modality in detecting satellite lesions as well as the most accurate in determining lesion sizes and concordances.

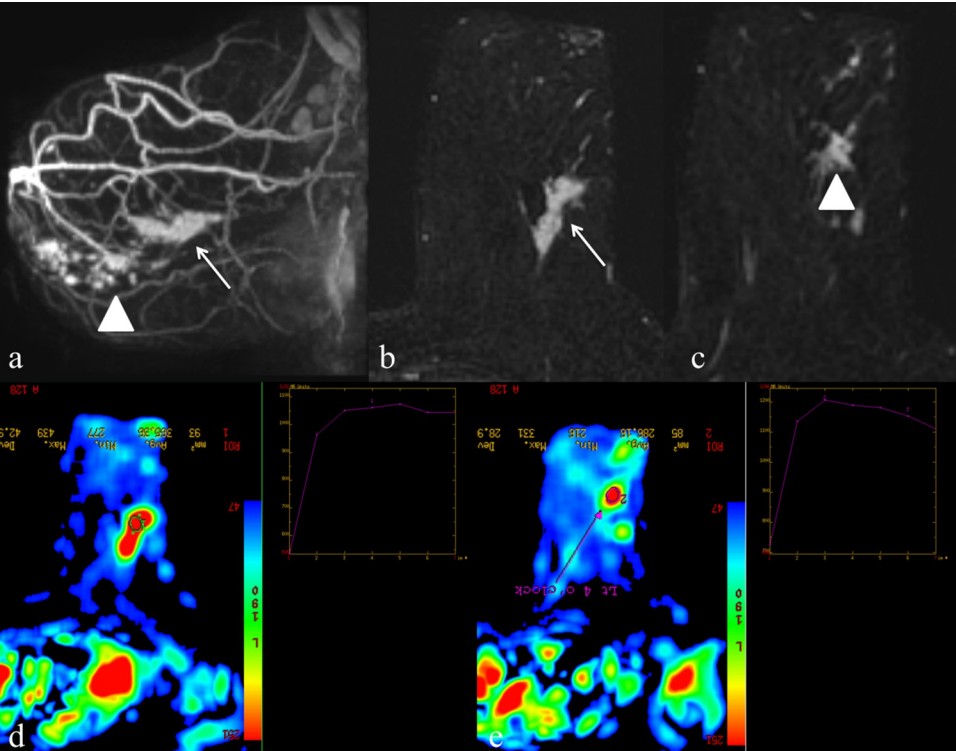

**Fig 5. A 49-year-old woman with invasive carcinoma who subsequently underwent mastectomy due to imaging discordance.** Breast MRI (A) Maximal intensity projection (MIP) shows the index tumour (thin arrow) and multifocal segmental clumped NME (HPE was DCIS) (arrowhead). (B, C) Axial subtracted post-contrast arterial phase 1 image shows index tumour (thin arrow) and multifocal lesion (arrowhead). (D) Colour coded intensity map of index tumour (red coloured area signifying the most intense enhancement) with type II kinetic curve (subset). (E) Colour coded intensity map of the multifocal lesion with Type III kinetic curves (subset).

The sensitivity of US alone in the detection of satellite lesions was 36.4%. The low sensitivity was due to the presence of many false-negative results. Our results closely correlate to the sensitivity quoted in a study which showed sensitivity of between 22.2% and 43.8% [37]. A study by Rabasco et al noted that 2D MMG alone missed 14 of 22 multifocal-multicentric lesions [34] leading to low sensitivity. Reasons for missed satellite lesions on MMG in several studies included dense breast parenchyma, technical errors, and misinterpretation of suspicious findings [37, 38]. On the other hand, MRI detected 8 out of the 11 satellite lesions, resulting in a sensitivity of 72.7%. The 3 satellite lesions that were not detected on MRI were adjacent to the index tumours and were removed together during the initial surgery. These lesions were later confirmed to be DCIS on HPE. Hence, no additional surgeries were needed in these patients. Our MRI findings correlate with Malur et al [39] and Hlawatsch et al [38] that reported 66.7% and 81% sensitivity of MRI in detecting satellite lesions respectively.

A total of 25 additional lesions were detected on MRI alone, out of which 16.7% were proven to be malignant on HPE, correlating with the figure quoted in a meta-analysis by Houssami et al of 16% additional disease detected by breast MRI [40]. Identification of these additional lesions pre-operatively will lower the recurrence rates and allow removal in the same surgical setting without the need for secondary surgery [28]. We also found that the percentages of patients requiring secondary surgeries for tumour resections due to positive margins were halved in the MRI group compared to the conventional group (4.8% on MRI versus 9.5% on conventional imaging) corresponding with findings of a study by Obdeijn et al which

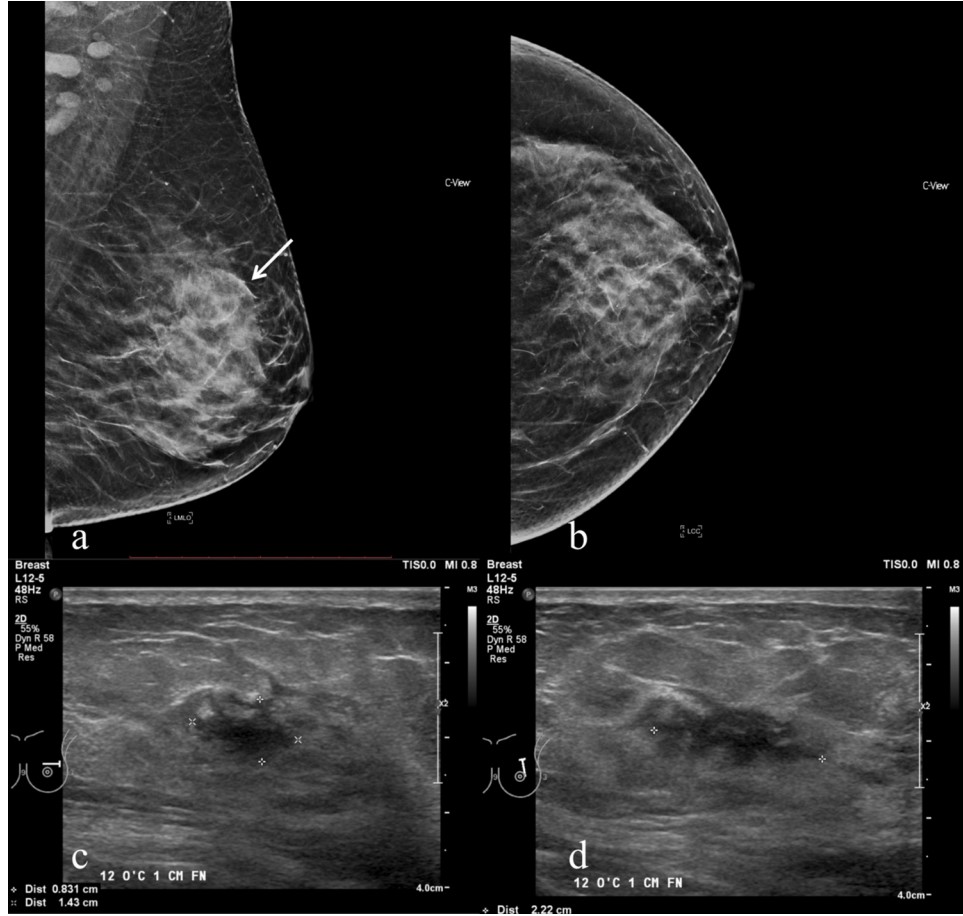

**Fig 6. 49-year-old woman with left breast invasive carcinoma who was deemed eligible for IORT based on conventional imaging but the tumour was larger than 30mm on MRI (done 4 days apart).** Left MMG in (A) MLO and (B) CC view showing an indistinct high-density mass (thin arrow) in the left upper central region in a background of dense breast parenchyma BIRADS. (C) Corresponding ultrasound of the left breast in (C) transverse and (D) longitudinal views show irregular hypoechoic lesion with angular margins. The largest dimension on conventional imaging was 22mm.

showed that the rates of reoperations for involved margins were approximately halved when comparing the MRI group with the control group (18.9% versus 37.4% respectively) [41]. The underestimation of the tumour size on MRI was the cause of positive margins in these patients. Recurrence rate was quoted at 5–25% in patients with positive margins [42] with the status of surgical margin being the most important factor for local recurrence prediction.

Pre-operative MRI findings impacted the surgical management of our patients. IORT was cancelled in 5 patients (11.9%) due to the presence of multifocal-multicentric diseases, tumour sizes larger than 30mm on MRI (which is incompatible with IORT) and presence of axillary lymphadenopathy, correlating with figures quoted in other studies which ranged between 5% and 12.5% [6, 17, 34]. True positive MRI findings in our study prompted the conversion of BCS to mastectomy in 3 patients (7.1% of total patients in MRI group). The subsequent surgical HPE of DCIS diseases in these patients deemed these conversions appropriate and our conversion rate is in accordance with those reported by Luciana Karla et al, 4.9% and Plana et al, 8.3% [22, 31]. Of the remaining 2 patients, one was subjected to NAC before the decision on subsequent surgery and the other was subjected to additional EBRT as a result of axillary lymph nodes involvement detected on MRI.

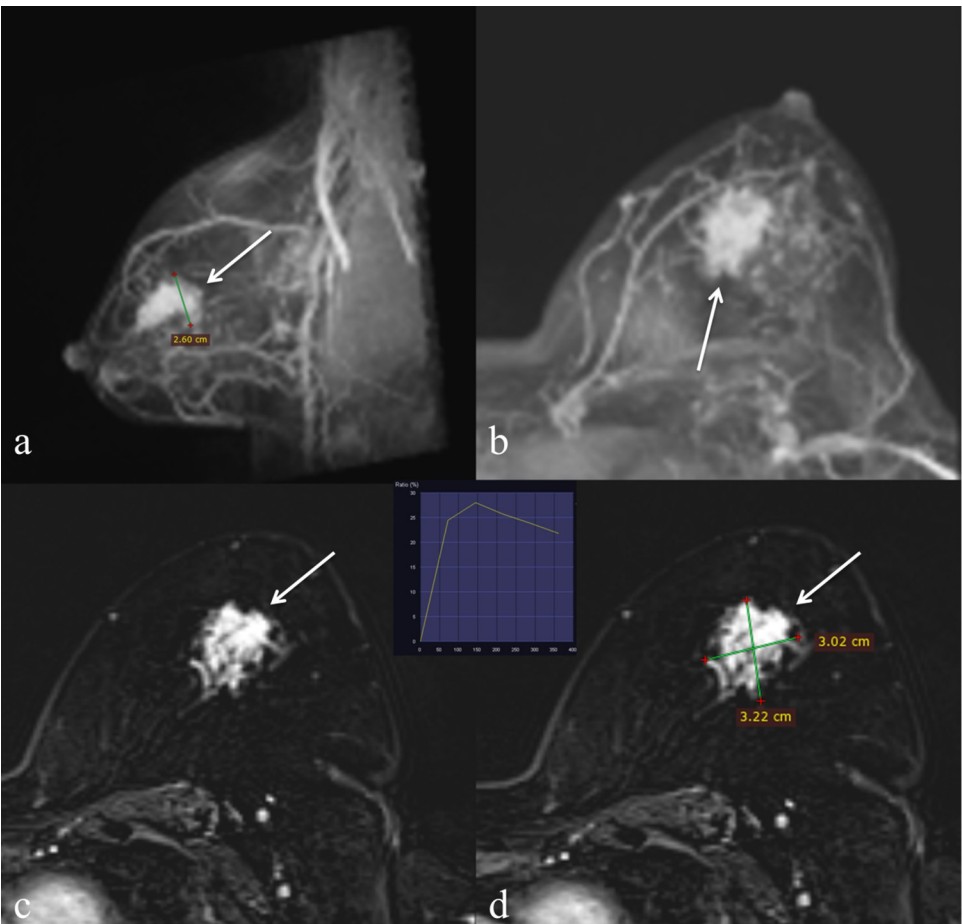

**Fig 7. 49-year-old woman with left breast invasive carcinoma who was deemed eligible for IORT based on conventional imaging but the tumour was larger than 30 mm on MRI (done 4 days apart).** Breast MRI Images maximal intensity projection (MIP) in (A) sagittal and (B) axial projections showing the index tumour (thin arrow). (C, D) Axial DCE phase 1 showing index tumour (thin arrow) with a maximal diameter (AP) of 32mm on MRI.

For lesion size, our study showed that US had the lowest concordance rate (57.6%) amongst all modalities, underestimating lesion sizes in 38.3% of patients (mean underestimation of 4.5mm). This was similarly noted in several studies which showed significant size

**Table 3. True positive (TP), true negative (TN), false positive (FP) and false negative (FN) in US and MRI.**

|  |  |  | Satellite lesion in pathology specimen | |
|---|---|---|---|---|
|  |  |  | Yes | No |
| **Satellite lesion in US** |  | Yes | 4 (TP)* | 6 (FP) |
|  |  | No | 7 (FN) | 43 (TN) |
|  |  |  | Satellite lesion in pathology specimen | |
|  |  |  | Yes | No |
| **Satellite lesion in MRI** |  | Yes | 8 (TP)* | 9 (FP) |
|  |  | No | 3 (FN) | 19 (TN) |

*TP lesions seen in US and MRI were multifocal lesions adjacent to the index tumour, which were collectively <30mm. These TP lesions were HPE confirmed DCIS post-surgical excision.

Table 4.  Concordances of tumour size by modality versus pathology.

|  | Group 1 (≤ 20mm) | | | Overall | | |
|---|---|---|---|---|---|---|
|  | US | MMG | MRI | US | MMG | MRI |
| N |  |  |  | 60 | 29 | 39 |
| Concordant | 33 | 12 | 20 | 34 (56.7%) | 20 (69%) | 27 (69.2%) |
| Underestimated | 23 | 3 | 3 | 23 (38.3%) | 4 (13.8%) | 5 (12.8%) |
| Overestimated | 1 | 1 | 1 | 3 (5.0%) | 5 (17.2%) | 7 (18.0%) |
|  | Group 2 (> 20mm) | | |  |  |  |
|  | US | MMG | MRI |  |  |  |
| Concordant | 1 | 8 | 7 |  |  |  |
| Underestimated | 0 | 1 | 2 |  |  |  |
| Overestimated | 2 | 4 | 6 |  |  |  |

*Data is presented in absolute number of patients unless specified otherwise.

underestimation by US [20, 43] with a mean difference of 2.5 mm [44]. Lesion sizes were underestimated in 13.8% of patients on MMG, while underestimation on MRI was seen in 12.8% of patients. Concordance rates of lesion size on MMG and MRI in this study were 69% and 69.2% respectively, which correlates with a concordance rate of 74.6% on MRI reported by Yoo EY et al [21]. Mean size differences in this study were 0.8 mm (underestimated on MRI) and 1.1 mm (overestimated on MMG) which were similar to figures quoted in a similar study which showed a mean difference of 0.6 mm (overestimated by MMG) [45]. In this study, we found that size discordances were seen in larger lesions (sizes > 20 mm) and lesions that are ER/PR negative. ER/PR negative patients made up less than 5 of this patient population. Other studies showed discordant imaging-pathological measurements as a result of tumour type (particularly invasive lobular carcinoma due to its histologic characteristics of diffuse infiltrative growth pattern and peritumoural satellite foci) [46, 47], ER negativity, lymphovascular invasion [21] and larger lesions exceeding US transducer size [48] leading to lesion size underestimation on US. Our study showed no statistically significant correlation between breast density and tumour size discordance, as reported in a study by Abu-Sinn et al [49]. Based on these findings, we concluded that MRI was the best modality in determining tumour size, which was in agreement with the findings of many studies [21, 23, 24, 43].

Although US had the lowest sensitivity in detection of satellite lesions and had significant size underestimation compared to pathological findings, US was still an useful tool in

Table 5.  Comparison of tumour size by modality versus pathology.

|  | Group 1 (≤ 20mm) | | | Overall | | |
|---|---|---|---|---|---|---|
|  | US | MMG | MRI | US | MMG | MRI |
| N (%) | 57 (95%) | 16 (55.2%) | 24 (61.5%) | 60 | 29 | 39 |
| Mean Difference* | -5.1 ± 1.26 | -0.94 ± 1.49 | -0.67 ± 1.07 | -4.5 ± 1.25 | 1.07 ± 1.38 | -0.77 ± 1.71 |
| P value | <0.001 | 0.539 | 0.540 | 0.001 | 0.446 | 0.655 |
|  | Group 2 (> 20mm) | | |  |  |  |
|  | US | MMG | MRI |  |  |  |
| N (%) | 3 (5%) | 13 (44.8%) | 15 (38.5%) |  |  |  |
| Mean Difference* | 23.3 ± 1.45 | 3.53 ± 2.37 | -0.93 ± 4.20 |  |  |  |
| P-value | 0.350 | 0.162 | 0.827 |  |  |  |

*Positive value = overestimated, Negative value = underestimated

P-value ≤ 0.05 is taken as significant.

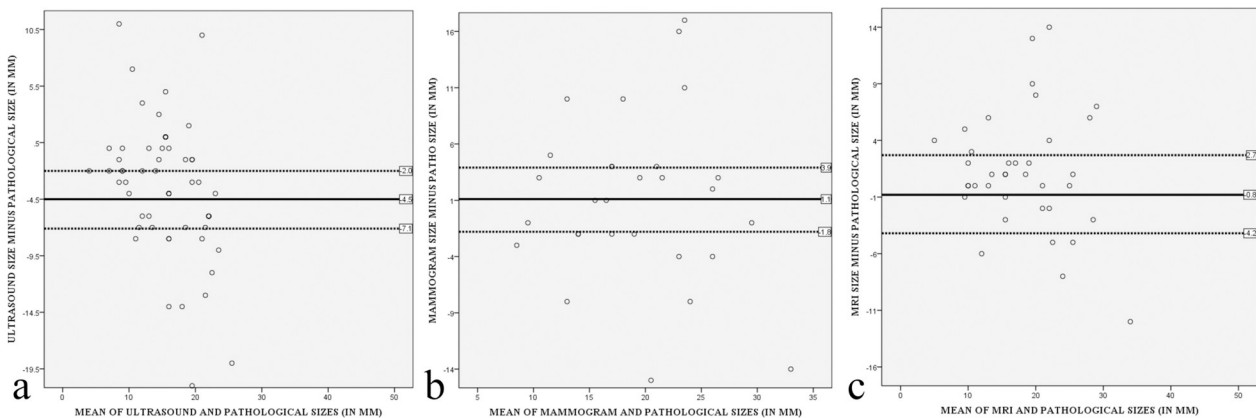

**Fig 8. Bland-Altman plots of imaging (US, MMG, and MRI) and pathological size differences versus mean size differences.**

decision-making of surgery and IORT. Studies have shown that US as an adjunct to MMG improves breast cancer diagnostic yield especially in women with dense breast [50, 51]. Adding US to MMG was found to be significantly more sensitive in cancer detection compared to MMG alone with no significant difference in specificity [52]. Most of the initial breast biopsy in our patients are via US guided and all the additional lesions detected on MRI in this study were reassess by second look US and some underwent US guided biopsies.

Although we observed an increase in the mean 'wait time' of 10 days in the MRI group compared to the conventional imaging group, this was shorter than the delay quoted by Bleicher et al [27] and Zhang et al [30] which showed an average of 22.4-day delay in their patients as a result of MRI due to the additional imaging and biopsies preformed post breast MRI. The delay observed in our study was due to 1–2 week average MRI appointment waiting time in our centre. Li Y et al [53] noted no significant decrease in patients' disease free survival with delay of more than 90 day, However, on the other hand Richards et al [54] showed a 12% lower 5-year survival with delay of more than 90 days. The delay encountered in our study likely have no impact on patients' disease free survival. The majority (85%) of patients in this population also were of Luminal A subtype (ER/PR positive, HER2-negative) which has demonstrated slower growth and lower grade in past literature, thus strengthening this belief [55].

Small sample size was the major limiting factor in this study. Although there were lesser patients with involved margins in the MRI group, there was no statistical significance which may be due to small cohort. There were only limited numbers of patients with ER and PR negativity (4 in total) for a more representative analysis. Thus it is difficult to discuss the role of ER/PR negativity in the selection of patients for preoperative MRI in this population.

## Conclusion

In our study, MRI was proven to be a useful adjunct to conventional imaging in patients with early breast cancer due to its high sensitivity, negative predictive value, high imaging-pathological size concordance, and ability to detect satellite lesions which may be missed by conventional imaging. The use of MRI reduced the risk of involved margins by two-folds. However, a larger sample size would be required to determine its significance. MRI also impacted the surgical management in a fraction of our patients.

Although our findings concluded that MRI had a definitive role as an adjunct to conventional imaging in the pre-operative assessment of patients planning for BCS and IORT, the value of routine use of pre-operative MRI in the work-up of patients for BCS and IORT

remains controversial. For now, until long-term data on clinical outcomes and cost-effectiveness are available, it may be more beneficial to use pre-operative MRI in selective patients.

## Supporting information

**S1 Data.**
(XLSX)

## Acknowledgments

Special thanks to Dr. Joanne Aisha Mosiun, Staff Nurse Zarinah, and University Malaya Research Imaging Centre (UMRIC) for the support throughout the entire study period.

## Author Contributions

**Conceptualization:** Mee Hoong See, Nur Aishah Mohd Taib, Kartini Rahmat.

**Data curation:** Wai Keong Cheah.

**Formal analysis:** Wai Keong Cheah, Mohammad Nazri Md Shah.

**Funding acquisition:** Wai Yee Chan, Kartini Rahmat.

**Investigation:** Wai Yee Chan, Wai Keong Cheah, Marlina Tanty Ramli Hamid.

**Methodology:** Wai Yee Chan, Wai Keong Cheah, Marlina Tanty Ramli Hamid, Mohammad Nazri Md Shah, Kartini Rahmat.

**Project administration:** Kartini Rahmat.

**Supervision:** Wai Yee Chan, Marlina Tanty Ramli Hamid.

**Validation:** Mohammad Nazri Md Shah.

**Writing – original draft:** Wai Keong Cheah.

**Writing – review & editing:** Wai Yee Chan, Marlina Tanty Ramli Hamid, Farhana Fadzli, Shaleen Kaur, Mee Hoong See, Nur Aishah Mohd Taib, Kartini Rahmat.

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
