## [Decision Letter · Decision Letter 0]

17 Feb 2022

PONE-D-21-32579IMPACT OF PREOPERATIVE MRI ON SURGERY AND ELIGIBILITY FOR INTRAOPERATIVE RADIOTHERAPY (IORT) IN EARLY BREAST CANCERPLOS ONE

Dear Dr. Ramli Hamid,

Thank you for submitting your manuscript to PLOS ONE. After careful consideration, we feel that it has merit but does not fully meet PLOS ONE’s publication criteria as it currently stands. Therefore, we invite you to submit a revised version of the manuscript that addresses the points raised during the review process.

We look forward to receiving your revised manuscript.

Kind regards,

Rubens Chojniak, M.D., Ph.D.

Academic Editor

PLOS ONE

Journal Requirements:

Additional Editor Comments:

This is a study that addresses an important and current issue, the indication of preoperative magnetic resonance imaging in breast cancer and its eventual impact on eligibility for intraoperative radiotherapy.

Secondarily, the study evaluated the accuracy of different imaging modalities in defining the dimensions of breast cancer using the surgical specimen as a reference standard, this is a data already well evaluated in studies with a larger number of patients and greater methodological rigor.

The reviewers raised important questions that I ask the authors to address and I emphasize the importance of describing the criteria for indicating MRI in this study in detail and the characteristics of patients in both study groups, with and without preoperative MR. Some pathological charateristics are described but there are some known criteria that favor the use of preoperative MRI due to the chance of impact on the conduct, I can mention Triple negative/HER-2/luminal B, High grade DCIS, Invasive lobular histology, dense breasts and young patients` tumors. Some of these characteristics of the population and subgroups were not provided and may influence the results.

Reviewers' comments:

Reviewer's Responses to Questions

**Comments to the Author**

1. Is the manuscript technically sound, and do the data support the conclusions?

Reviewer #1: Yes

Reviewer #2: Yes

2. Has the statistical analysis been performed appropriately and rigorously? 

Reviewer #1: Yes

Reviewer #2: Yes

3. Have the authors made all data underlying the findings in their manuscript fully available?

Reviewer #1: Yes

Reviewer #2: Yes

4. Is the manuscript presented in an intelligible fashion and written in standard English?

Reviewer #1: Yes

Reviewer #2: Yes

5. Review Comments to the Author

Reviewer #1: The manuscript titled 'Impact of preoperative MRI on surgery and eligibility for intraoperative radiotherapy

(IORT) in early breast cancer' by Chan et al examined the impact of pre-operative MRI on decision-making of surgery and IORT for 63 breast cancer patients randomized into two groups. Their findings are interesting and will have a positive effect on clinical practice if published. There are some minor concerns regarding the manuscript.

1. Since MRI and mammogram have better correlation with pathological findings, is ultrasound still an useful tool in decision-making of surgery and IORT? Please discuss the issue in the discussion section.

2. Since sample size is small in this study, statistical power of the analysis is limited. Does the authors have a plan to collect more patients to increase their sample size and make stronger conclusion?

3. BCS was not explained the first time when the abbreviation appeared in the 'abstract' although it was explained in the 'Introduction'.

Reviewer #2: Dear Authors,

About the article "IMPACT OF PREOPERATIVE MRI ON SURGERY AND ELIGIBILITY FOR INTRAOPERATIVE RADIOTHERAPY (IORT) IN EARLY BREAST CANCER", please consider the following aspects:

1- In the title, avoid using abbreviations, unless you also indicate their meaning. For example, MRI (Magnetic Resonance Imaging);

2- In the abstract, consider including the abbreviation of some keywords such as magnetic resonance imaging, ultrasound and mammography, and use this standardization throughout the text, since sometimes these words are abbreviated and sometimes not throughout the text. Or don’t abbreviate them in the abstract and start using the abbreviations, after properly signaling them, starting the introduction.

3- Match the objective of the abstract with that of the introduction. There is an objetive a correlation between the “waiting time” and the study groups in relation to the methods to which they are submitted, not informed to the reader in the abstract of the article;

4- Regarding the study population, consider whether it is interesting to know how many patients were excluded from the analysis, perhaps in relation to the number of MRI scans done in the same period. Consider using a flow diagram.

5- Regarding the study population, consider inserting in the flow diagram (Figure 3) all the patients elected for the study and not only those who underwent MRI, in order to better systematize the data.

6- In the methods, please provide more information regarding tumor characteristics and the identification of satellite and metastatic lesions (multifocal-multicentric disease), reported in the results, such as those included in tables 1 and 2.

7- In the methods, clearly describe the variables related to patients and tumor characteristics, such as breast parenchyma density, as these are potential confounding factors and effect modifiers.

8- Consider standardizing breast lesions dimensions in "mm or cm" in the methods and results.

9- In the results, consider performing the following analyses in order to rule out possible biases regarding the evaluation of tumor size in relation to the different methods:

- breast density x study groups (MMG and US x MRI).

10- In the discussion, an interesting reflection would be the impact that the result of this work brings to clinical practice.

Other observations:

Page 2, line 44 - Include the term referring to the abbreviation MRI.

Page 2, line 51 – Include the term referring to the abbreviation BCS.

Page 3, line 80 - Include the expression referring to the abbreviation UMMC.

Page 3, lines 80 to 82 - What is the bibliographic reference based for the classification of IORT indications in UMMC?

Page 4, line 85 - Leave only the abbreviations for mammography and ultrasound, since they were indicated on lines 68 and 69, respectively.

Page 4, line 96 and 97 - Leave only the abbreviation BCS.

6. PLOS authors have the option to publish the peer review history of their article (what does this mean?). If published, this will include your full peer review and any attached files.

Reviewer #1: No

Reviewer #2: No

---

## [Author Response · Author response to Decision Letter 0]

10 Apr 2022

To 

PLOS ONE Editorial Office 

Dear Sir/Dr,

Response to Reviewer’s Comments.

Ms. No.: PONE-D-21-32579

Title: IMPACT OF PREOPERATIVE MRI ON SURGERY AND ELIGIBILITY FOR INTRAOPERATIVE RADIOTHERAPY (IORT) IN EARLY BREAST CANCER

Thank you very much for your comments to the above referenced manuscript and for consideration for publication in your respected journal. We highly appreciate the reviewers’ comments and have revised the manuscript according to the reviewer’s recommendation (see point-by-point reply enclosed herewith). We thank the reviewers for their valuable suggestions in improving the paper. The substantive changes have been addressed below with regards to each reviewers questions. 

Below is the corrections/point-by-point reply to your comments.

Journal Requirements:

Authors’ reply: Thank you for the comments. We have followed the PLOS ONE’s style requirement and file naming as per template.

Authors’ reply: Thank you for the comments. The ethics statement is in the Materials and Methods section under Study Design subsection as highlighted. We have deleted the statement that was previously in the Acknowledgements section. 

Additional Editor Comments:

This is a study that addresses an important and current issue, the indication of preoperative magnetic resonance imaging in breast cancer and its eventual impact on eligibility for intraoperative radiotherapy.

Secondarily, the study evaluated the accuracy of different imaging modalities in defining the dimensions of breast cancer using the surgical specimen as a reference standard, this is a data already well evaluated in studies with a larger number of patients and greater methodological rigor.

The reviewers raised important questions that I ask the authors to address and I emphasize the importance of describing the criteria for indicating MRI in this study in detail and the characteristics of patients in both study groups, with and without preoperative MR. Some pathological charateristics are described but there are some known criteria that favor the use of preoperative MRI due to the chance of impact on the conduct, I can mention Triple negative/HER-2/luminal B, High grade DCIS, Invasive lobular histology, dense breasts and young patients` tumors. Some of these characteristics of the population and subgroups were not provided and may influence the results.

Reviewers' comments:

Reviewer's Responses to Questions

Comments to the Author

1. Is the manuscript technically sound, and do the data support the conclusions?

Reviewer #1: Yes

Reviewer #2: Yes

Authors’ reply: Thank you for the favourable response.

2. Has the statistical analysis been performed appropriately and rigorously?

Reviewer #1: Yes

Reviewer #2: Yes

Authors’ reply: Thank you for the favourable response.

3. Have the authors made all data underlying the findings in their manuscript fully available?

Reviewer #1: Yes

Reviewer #2: Yes

Authors’ reply: Thank you for the favourable response.

4. Is the manuscript presented in an intelligible fashion and written in standard English?

Reviewer #1: Yes

Reviewer #2: Yes

Authors’ reply: Thank you for the favourable response.

5. Review Comments to the Author

Reviewer #1: The manuscript titled 'Impact of preoperative MRI on surgery and eligibility for intraoperative radiotherapy

(IORT) in early breast cancer' by Chan et al examined the impact of pre-operative MRI on decision-making of surgery and IORT for 63 breast cancer patients randomized into two groups. Their findings are interesting and will have a positive effect on clinical practice if published. There are some minor concerns regarding the manuscript.

1. Since MRI and mammogram have better correlation with pathological findings, is ultrasound still an useful tool in decision-making of surgery and IORT? Please discuss the issue in the discussion section.

Authors’ reply: Thank you for your comments. Although MRI and MMG have better correlation with pathological findings, initial imaging assessment of early breast cancer patients for eligibility for BCS and IORT are done with MMG and US. Breast MRI is not a routine assessment tool for patients with breast cancer. Studies (1,2,3) have shown that US as an adjunct to MMG improves breast cancer diagnostic yield especially in women with dense breast. Adding US to MMG was found to be significantly more sensitive in cancer detection compared to MMG alone (80.8% vs 56.6%) with no significant difference in specificity. Most of the initial biopsy in patients presenting with suspicious breast related symptoms are via US guided. All the additional lesions detected on MRI in this study were reassess by second look US and some lesions underwent US guided biopsies. We have included this in the discussion section from line 400 – 405 of the manuscript. 

1. Corsetti V, Houssami N, Ferrari A, Ghirardi M, Bellarosa S, Angelini O, et al. Breast screening with ultrasound in women with mammography-negative dense breasts: evidence on incremental cancer detection and false positives, and associated cost. European journal of cancer. 2008;44(4):539-44.

2. McCavert M, O’Donnell M, Aroori S, Badger S, Sharif M, Crothers J, et al. Ultrasound is a useful adjunct to mammography in the assessment of breast tumours in all patients. International journal of clinical practice. 2009;63(11):1589-94. 

3. Yip CH, Pathy NB, Teo SH. A review of breast cancer research in Malaysia. Med J Malaysia. 2014;69(August):8–22.

2. Since sample size is small in this study, statistical power of the analysis is limited. Does the authors have a plan to collect more patients to increase their sample size and make stronger conclusion?

Authors’ reply: Thank you for your comments. Increment of the sample size is beyond the scope of the current study. 

3. BCS was not explained the first time when the abbreviation appeared in the 'abstract' although it was explained in the 'Introduction'.

Authors’ reply: Thank you for your comments. We have edited and included the full term/form of BCS which is Breast-Conserving Surgery in the abstract.

Reviewer #2: Dear Authors,

About the article "IMPACT OF PREOPERATIVE MRI ON SURGERY AND ELIGIBILITY FOR INTRAOPERATIVE RADIOTHERAPY (IORT) IN EARLY BREAST CANCER", please consider the following aspects:

1- In the title, avoid using abbreviations, unless you also indicate their meaning. For example, MRI (Magnetic Resonance Imaging);

Authors’ reply: Thank you for your comments. We have edited and included only the full term/form of MRI which is Magnetic Resonance Imaging in the title and also remove IORT abbreviation in the title.

2- In the abstract, consider including the abbreviation of some keywords such as magnetic resonance imaging, ultrasound and mammography, and use this standardization throughout the text, since sometimes these words are abbreviated and sometimes not throughout the text. Or don’t abbreviate them in the abstract and start using the abbreviations, after properly signaling them, starting the introduction.

Authors’ reply: Thank you for your comments. We have standardized our manuscript with regards to the use of abbreviations. We have used the full term with abbreviation in the abstract and started using these abbreviations from the introduction section. 

3- Match the objective of the abstract with that of the introduction. There is an objetive a correlation between the “waiting time” and the study groups in relation to the methods to which they are submitted, not informed to the reader in the abstract of the article;

Authors’ reply: Thank you for your comments. We have included the results of the wait time for surgical management in the abstract in line 60 and 61 as per below:

“Significant increase in the mean ‘waiting time’ were seen in the MRI group (34.1 days) compared to the conventional imaging group (24.4 days)” 

The results and the discussion on the “waiting time” were included in the Results section under the subheading of Patient population demographics and breast MRI findings from line 200 to 204 and in the Discussion section from line 408-416.

4- Regarding the study population, consider whether it is interesting to know how many patients were excluded from the analysis, perhaps in relation to the number of MRI scans done in the same period. Consider using a flow diagram.

Authors’ reply: Thank you for your comments. The inclusion criteria for this study are all patients with early breast cancer that are deemed to be eligible for BCS and IORT on conventional imaging. This include patients with unifocal disease with breast tumour size ≤ 3cm with no lymph node involvement or metastasis. Patients requiring neoadjuvant chemotherapy and those with MRI contraindication were excluded. These were mentioned under patient selection subheading from line 136 to 141. Unfortunately however, we do not have the data on the no of patients excluded from the analysis in relation to the total no of MRI done in the same period of time. 

5- Regarding the study population, consider inserting in the flow diagram (Figure 3) all the patients elected for the study and not only those who underwent MRI, in order to better systematize the data.

Authors’ reply: Thank you for your comments. Figure 3 flowchart has already included the outcome for all 63 early breast cancer patients deemed eligible for this study. The patients were divided into two arms: those with conventional imaging (MMG and US) (n=21) alone and those with MRI as adjunct to conventional imaging (n=42).

6- In the methods, please provide more information regarding tumour characteristics and the identification of satellite and metastatic lesions (multifocal-multicentric disease), reported in the results, such as those included in tables 1 and 2.

Authors’ reply: Thank you for your comments. This study included 63 patients with early breast cancer (unifocal invasive ductal carcinoma with tumour size ≤ 3cm with no lymph node involvement or metastasis) that are deemed to be eligible for BCS and IORT on conventional imaging. Except for patients requiring neoadjuvant chemotherapy and those with MRI contraindication, the rest of the patients included in the study were offered MRI. This was mentioned under patient selection subheading from line 136 to 141. Table 1 summarised the detailed characteristics of the tumour included in our study. 

Additional lesions detected on MRI were classified according to ACR BI-RADS 2015 lexicon into benign and suspicious masses, and non-mass enhancement (NME). For masses, these include assessment of lesions’ shape, margin and internal enhancement characteristic. NMEs were assessed according to the distribution and internal enhancement pattern. Multifocal disease is the presence of another focus/foci of cancer (satellite lesion) in the same quadrant of the index tumour , which collectively measures <3cm, whilst multicentric disease means presence of cancer foci in more than one quadrant of the breast. The management of these additional lesions were decided in a multidisciplinary team discussion with the surgical team. This was mentioned under data collection and analysis subheading from line 159 to 174. 

7- In the methods, clearly describe the variables related to patients and tumor characteristics, such as breast parenchyma density, as these are potential confounding factors and effect modifiers.

Authors’ reply: Thank you for your comments. The table below summarizes the risk factors attributable to breast cancer in all the 63 patients involved in this our study. Many of these data were not documented in detail in the Electronic Medical Records (EMR)

Risk factors of patients with breast cancer (N=63)

Risk factors N %

Early menarche (age <12)

Late onset menopause (age ≥ 55)

Smoking

Alcohol intake

Hormonal therapy (HRT, OCP)

Para

 Nulliparous

 Multiparous

Breastfeeding

Family history of breast cancer 6/63

6/51*

2/63

6/63

21/63

4

59

54/59**

17/63 9.5

11.8

3.2

9.5

33.3

6.3

93.7

91.5

27.0

*51 patients had attained menopause, 12 were still premenopausal. 

**59 patients were multiparous, 4 patients were nulliparous.

As for breast parenchymal density, out of the 63 patients, the majority of patients in this study have BIRADS breast density B. 5 (7.9%) have extremely dense breast which would lower the sensitivity of mammography. Seven (7) tumours were missed on mammography but detected on supplementary US which were performed due to either heterogeneously or extremely dense breast parenchyma. The table below summarizes the breast density of all patients involved in this study. 

Mammographic breast density (N=63)

Breast Density N %

 A

 B

 C

 D 9

26

23

5 14.3

41.3

36.5

7.9

These were mentioned in results section under subheading patient population demographics and breast MRI findings from line 200 to 204

In view of the limited number and incomplete EMR documentation, the assessment and statistical analysis of variables related to patients eg breast parenchyma density, as potential confounding factors and effect modifiers with tumour characteristics are beyond the scope of this study. 

8- Consider standardizing breast lesions dimensions in "mm or cm" in the methods and results.

Authors’ reply: Thank you for your comments. We have standardized our manuscript and have used “mm” instead of “cm” throughout.

9- In the results, consider performing the following analyses in order to rule out possible biases regarding the evaluation of tumor size in relation to the different methods:

- breast density x study groups (MMG and US x MRI).

Authors’ reply: Thank you for your comments. Unfortunately, these analyses are beyond the scope of our study 

10- In the discussion, an interesting reflection would be the impact that the result of this work brings to clinical practice. 

Authors’ reply: Thank you for your comments. We have included the impact of the results of this study in the conclusion section. 

Other observations:

Page 2, line 44 - Include the term referring to the abbreviation MRI.

Authors’ reply: Thank you for your comments. We have edited and included the full term/form of MRI which is Magnetic Resonance Imaging in the abstract.

Page 2, line 51 – Include the term referring to the abbreviation BCS.

Authors’ reply: Thank you for your comments. We have edited and included the full term/form of BCS which is Breast-Conserving Surgery in the abstract.

Page 3, line 80 - Include the expression referring to the abbreviation UMMC.

Authors’ reply: Thank you for your comments. We have edited and included the full term/form of UMMC which is University Malaya Medical Centre in the introduction section. 

Page 3, lines 80 to 82 - What is the bibliographic reference based for the classification of IORT indications in UMMC?

Authors’ reply: Thank you for your comments. We have added the he bibliographic reference for IORT classifications in UMMC as per below. 

Esposito E, Douek M. Update on intraoperative radiotherapy: new challenges and issues. Ecancermedicalscience. 2018;12:793. Published 2018 Jan 10. doi:10.3332/ecancer.2018.793

Page 4, line 85 - Leave only the abbreviations for mammography and ultrasound, since they were indicated on lines 68 and 69, respectively.

Authors’ reply: Thank you for your comments. We have standardized our manuscript with regards to the use of abbreviations. We have used the full term with abbreviation in the abstract and started using these abbreviations from the introduction section. 

Page 4, line 96 and 97 - Leave only the abbreviation BCS.

Authors’ reply: Thank you for your comments. We have standardized our manuscript with regards to the use of abbreviations. We have used the full term with abbreviation in the abstract and started using these abbreviations from the introduction section.

---

## [Decision Letter · Decision Letter 1]

19 May 2022

PONE-D-21-32579R1Impact of preoperative magnetic resonance imaging on surgery and eligibility for intraoperative radiotherapy in early breast cancerPLOS ONE

Dear Dr. Ramli Hamid,

Thank you for submitting your manuscript to PLOS ONE. After careful consideration, we feel that it has merit but does not fully meet PLOS ONE’s publication criteria as it currently stands. Therefore, we invite you to submit a revised version of the manuscript that addresses the points raised during the review process.

The manuscript has been assessed by two reviewers, and their comments are appended below.

While the reviewers report that most of their comments have been addressed, some concerns still remain.

Could you please revise the manuscript to carefully address the concerns raised?

We look forward to receiving your revised manuscript.

Kind regards,

Jamie Royle

Staff Editor

PLOS ONE

Journal Requirements:

Reviewers' comments:

Reviewer's Responses to Questions

**Comments to the Author**

1. If the authors have adequately addressed your comments raised in a previous round of review and you feel that this manuscript is now acceptable for publication, you may indicate that here to bypass the “Comments to the Author” section, enter your conflict of interest statement in the “Confidential to Editor” section, and submit your "Accept" recommendation.

Reviewer #1: All comments have been addressed

Reviewer #2: All comments have been addressed

2. Is the manuscript technically sound, and do the data support the conclusions?

Reviewer #1: Yes

Reviewer #2: Yes

3. Has the statistical analysis been performed appropriately and rigorously? 

Reviewer #1: Yes

Reviewer #2: Yes

4. Have the authors made all data underlying the findings in their manuscript fully available?

Reviewer #1: Yes

Reviewer #2: Yes

5. Is the manuscript presented in an intelligible fashion and written in standard English?

Reviewer #1: Yes

Reviewer #2: Yes

6. Review Comments to the Author

Reviewer #1: The authors have adequately addressed my concerns. This revised version is acceptable for publication.

Reviewer #2: This manuscript examined the impact of preoperative MRI on surgery and IORT decision making for 63 breast cancer patients in two groups. Their findings are interesting and will have a positive effect on practice if published. Almost all comments were adequately answered. There are still some minor concerns raised additionally by the editor and not answered:

"I ask the authors to address and I emphasize the importance of describing the criteria for indicating MRI in this study in detail and the characteristics of patients in both study groups, with and without preoperative MR. Some pathological charateristics are described but there are some known criteria that favor the use of preoperative MRI due to the chance of impact on the conduct, I can mention Triple negative/HER-2/luminal B, High grade DCIS, Invasive lobular histology, dense breasts and young patients` tumors. Some of these characteristics of the population and subgroups were not provided and may influence the results."

7. PLOS authors have the option to publish the peer review history of their article (what does this mean?). If published, this will include your full peer review and any attached files.

Reviewer #1: No

Reviewer #2: No

---

## [Author Response · Author response to Decision Letter 1]

14 Jun 2022

To 

PLOS ONE Editorial Office 

Dear Sir/Dr,

Response to Reviewer’s Comments.

Ms. No.: PONE-D-21-32579

Title: IMPACT OF PREOPERATIVE MRI ON SURGERY AND ELIGIBILITY FOR INTRAOPERATIVE RADIOTHERAPY (IORT) IN EARLY BREAST CANCER

Thank you very much for your comments to the above referenced manuscript and for consideration for publication in your respected journal. We highly appreciate the reviewers’ comments and have revised the manuscript according to the reviewer’s recommendation (see point-by-point reply enclosed herewith). We thank the reviewers for their valuable suggestions in improving the paper. The substantive changes have been addressed below with regards to each reviewers questions. 

Below is the corrections/point-by-point reply to your comments.

Journal Requirements:

Additional Editor Comments:

The reviewers raised important questions that I ask the authors to address and I emphasize the importance of describing the criteria for indicating MRI in this study in detail and the characteristics of patients in both study groups, with and without preoperative MR. Some pathological charateristics are described but there are some known criteria that favor the use of preoperative MRI due to the chance of impact on the conduct, I can mention Triple negative/HER-2/luminal B, High grade DCIS, Invasive lobular histology, dense breasts and young patients` tumors. Some of these characteristics of the population and subgroups were not provided and may influence the results.

Author’s reply:

Thank you for your constructive comment. In this study we did observe patients who were ER-/PR- as well as HER-2 negative and included them in the results particularly in Table 1. We have now added Table 2 which lists the Molecular Subtypes more clearly. We also noted that the majority of patients were in the category of BIRADS density B. We did not look at invasive lobular histology or DCIS in these patients as our focus as mentioned in methodology was on patients who had invasive ductal carcinoma. However, we did detect DCIS in some of the satellite lesions that were excised and this was discussed in the Discussion section. In addition, majority of our patients were in the 40-60 ages range, therefore to improve readability of this manuscript, we did not emphasise the younger age group. I hope that this addresses your comments and concerns.

Below is Table 2, which is located just after Table 1 in the manuscript.

Table 2. Frequency of Molecular Subtypes in the study population

Subtype N %

Luminal A 46/54 85.1

Luminal B 5/54 9.2

HER2+ 2/54 3.9

Triple Negative Breast Cancer 1/54 1.8

*9 patients did not have complete hormonal information in order to fit the above subtypes

---

## [Decision Letter · Decision Letter 2]

12 Jul 2022

PONE-D-21-32579R2Impact of preoperative magnetic resonance imaging on surgery and eligibility for intraoperative radiotherapy in early breast cancerPLOS ONE

Dear Dr. Ramli Hamid,

Thank you for submitting your manuscript to PLOS ONE. After careful consideration, we feel that it has merit but does not fully meet PLOS ONE’s publication criteria as it currently stands. Therefore, we invite you to submit a revised version of the manuscript that addresses the points raised during the review process.

Please see the section "Additional Editor comments".

We look forward to receiving your revised manuscript.

Kind regards,

Thomas Tischer

Staff Editor

PLOS ONE

Journal Requirements:

Additional Editor Comments (if provided):Thank you for providing additional tumor/patient characteristics in Table 2. You were previously asked to to address and emphasize the importance of these data as they may influence the results. We would like to invite you to add a paragraph in your results or discussion section to address this concern.We noticed that you provide the patient list with data as supplementary file. This should be mentioned in the material and methods section.

Reviewers' comments:

Reviewer's Responses to Questions

**Comments to the Author**

1. If the authors have adequately addressed your comments raised in a previous round of review and you feel that this manuscript is now acceptable for publication, you may indicate that here to bypass the “Comments to the Author” section, enter your conflict of interest statement in the “Confidential to Editor” section, and submit your "Accept" recommendation.

Reviewer #1: All comments have been addressed

2. Is the manuscript technically sound, and do the data support the conclusions?

Reviewer #1: Yes

3. Has the statistical analysis been performed appropriately and rigorously? 

Reviewer #1: Yes

4. Have the authors made all data underlying the findings in their manuscript fully available?

Reviewer #1: Yes

5. Is the manuscript presented in an intelligible fashion and written in standard English?

Reviewer #1: Yes

6. Review Comments to the Author

Reviewer #1: (No Response)

7. PLOS authors have the option to publish the peer review history of their article (what does this mean?). If published, this will include your full peer review and any attached files.

Reviewer #1: No

---

## [Author Response · Author response to Decision Letter 2]

15 Aug 2022

To 

PLOS ONE Editorial Office 

Dear Sir/Dr,

Response to Reviewer’s Comments.

Ms. No.: PONE-D-21-32579

Title: IMPACT OF PREOPERATIVE MRI ON SURGERY AND ELIGIBILITY FOR INTRAOPERATIVE RADIOTHERAPY (IORT) IN EARLY BREAST CANCER

Thank you very much for your comments to the above referenced manuscript and for consideration for publication in your respected journal. We highly appreciate the reviewers’ comments and have revised the manuscript according to the reviewer’s recommendation (see point-by-point reply enclosed herewith). We thank the reviewers for their valuable suggestions in improving the paper. The substantive changes have been addressed below with regards to each reviewers questions. 

Below is the corrections/point-by-point reply to your comments.

Journal Requirements:

Additional Editor Comments:

Thank you for providing additional tumor/patient characteristics in Table 2. You were previously asked to address and emphasize the importance of these data as they may influence the results. We would like to invite you to add a paragraph in your results or discussion section to address this concern.

Author’s reply:

Thank you Editor for your constructive comments. We have now added several lines in the discussion to allude to the data in Table 2: 

Line 394, 419 and 425. 

I hope that these will be sufficient to address your concerns which we much appreciate. 

We noticed that you provide the patient list with data as supplementary file. This should be mentioned in the material and methods section.

Author’s reply:

Thank you for your comment. We have now added a sentence within Materials and Methods: Data Collection and Analysis; Line 171.

Below is the response to the previous revision that we had done:

The reviewers raised important questions that I ask the authors to address and I emphasize the importance of describing the criteria for indicating MRI in this study in detail and the characteristics of patients in both study groups, with and without preoperative MR. Some pathological charateristics are described but there are some known criteria that favor the use of preoperative MRI due to the chance of impact on the conduct, I can mention Triple negative/HER-2/luminal B, High grade DCIS, Invasive lobular histology, dense breasts and young patients` tumors. Some of these characteristics of the population and subgroups were not provided and may influence the results.

Author’s reply:

Thank you for your constructive comment. In this study we did observe patients who were ER-/PR- as well as HER-2 negative and included them in the results particularly in Table 1. We have now added Table 2 which lists the Molecular Subtypes more clearly. We also noted that the majority of patients were in the category of BIRADS density B. We did not look at invasive lobular histology or DCIS in these patients as our focus as mentioned in methodology was on patients who had invasive ductal carcinoma. However, we did detect DCIS in some of the satellite lesions that were excised and this was discussed in the Discussion section. In addition, majority of our patients were in the 40-60 ages range, therefore to improve readability of this manuscript, we did not emphasise the younger age group. I hope that this addresses your comments and concerns.

Below is Table 2, which is located just after Table 1 in the manuscript.

Table 2. Frequency of Molecular Subtypes in the study population

Subtype N %

Luminal A 46/54 85.1

Luminal B 5/54 9.2

HER2+ 2/54 3.9

Triple Negative Breast Cancer 1/54 1.8

*9 patients did not have complete hormonal information in order to fit the above subtypes

---

## [Editor Report · Decision Letter 3]

28 Aug 2022

Impact of preoperative magnetic resonance imaging on surgery and eligibility for intraoperative radiotherapy in early breast cancer

PONE-D-21-32579R3

Dear Dr. Ramli Hamid,

We’re pleased to inform you that your manuscript has been judged scientifically suitable for publication and will be formally accepted for publication once it meets all outstanding technical requirements.

Kind regards,

George Vousden

Staff Editor

PLOS ONE
---

## [Editor Report · Acceptance letter]

6 Oct 2022

PONE-D-21-32579R3 

Impact of preoperative magnetic resonance imaging on surgery and eligibility for intraoperative radiotherapy in early breast cancer 

Dear Dr. Ramli Hamid:

I'm pleased to inform you that your manuscript has been deemed suitable for publication in PLOS ONE. Congratulations! Your manuscript is now with our production department. 

Kind regards, 

on behalf of

Dr. George Vousden 

Staff Editor

PLOS ONE